# OpenReview forum: "What Would Jiminy Cricket Do? Towards Agents That Behave Morally"
_NeurIPS.cc/2021/Track/Datasets_and_Benchmarks/Round2 — NeurIPS 2021 Datasets and Benchmarks Track (Round 2)_

### Official Review · Reviewer_Pke5 · 2021-09-03
**A substantial dataset for a useful purpose**

**Rating:** 7
**Confidence:** 3
**Clarity:** The paper is clearly written.

**Strengths:**

The paper addresses a relevant problem, since text-based games are a useful environment for AI research but the games' built-in rewards (i.e. game completion) are not necessarily aligned with morals. This dataset will be useful to leverage such games as environments for experimentation while creating morally-aware agents, and to evaluate how such agents perform with respect to morality.

The presented dataset represents a very substantial amount of work - a large number of actions have been annotated in a very detailed way by diving into 70s-80s source code.

The models presented in the paper and evaluated against the dataset are interesting not only for their own sake, but because they give an idea of what one can expect to achieve (i.e., the provided data and visualizations implicitly answer some natural questions like how much immorality one typically needs to finish a game, etc.)

**Weaknesses:**

While the presented models reduce immorality with respect to the baseline, they still have a long way to go to even approach the human expert. I do not consider this to be an important weakness as the main contribution here is the dataset, with the models just serving as a way to show how the dataset can be used and a starting point.

**Additional Feedback:**

While this does not affect the scientific value of the contribution, I think there is a natural question that the reader may ask themselves (at least, I did) and should be answered in the paper: why Infocom games? These are games written in a quite ancient programming language, and their copyright is unclear as they are abandonware (as the authors rightly point out in the supplementary material). Wouldn't it have been easier to use more modern games, written in more modern languages like Inform, and which are often open-source (or permission could be seeked from the authors when they aren't)? I would like to know (and I suppose, many interested readers would) what the reason for this choice is: is it that modern games tend to be shorter or more linear than Infocom games?

In any research dealing with language, we should specify what language we are using - the paper should explicitly say somewhere that this is a dataset of English games and all the models operate on English.

I spotted a couple of minor typos/inconsistencies:

l. 74: markov -> Markov

l. 116-117: "For instance, in Figure 1 an agent decides to take an object from a non-player character, who resists the attempted theft" -> according to the figure, the NPC had been killed, so they could not resist the theft.

**Correctness:**

The scientific content of the paper is sound as far as I can see. I only have a minor comment about the claim that "people have an innate sense of basics of right and wrong that guides their actions in everyday life". While I am no expert in these matters, I don't think one can categorically claim that there is such a thing as an innate (as opposed to learned) morality; my understanding is that that is not obvious and subject to debate. If the authors want to make this kind of claim, it would require some justification (references, etc.)

The paper should also at least mention the fact that morality is not universal (actions that are considered good in a given culture/country/religion are not necessarily so in a different one) and provide some data about the annotators' cultural background.

**Documentation:**

The dataset is available in GitHub and it is well documented in the supplementary material, which notably includes considerations about potential licensing problems and a datasheet.

**Ethics:**

The paper itself is concerned with training ethical agents and evaluating the degree of morality of agents. I see no ethical concerns with this contribution.

**Relation To Prior Work:**

The paper does a good job of putting the contribution in relation to previous work.

**Summary And Contributions:**

This paper presents a benchmark to evaluate text-based game AI agents in terms of moral behavior, rather than just focusing in reward functions that can go against morality, with the purpose of making the simulated environments provided by such games useful to develop ethical agents. For this purpose, the authors have annotated 25 classic Infocom text adventure games according to the morality of each relevant action. The paper also presents approaches that use pretrained language models (which can, to a certain extent, assess whether the actions described in a piece of text are moral or not) to condition the agent into a better moral behavior, showing that these approaches indeed reduce immoral actions without reducing the agent's performance in the game.

---

> ### Author Response · Authors · 2021-09-30
> **Response to Reviewer Pke5**
>
> Thank you for your careful analysis of our work. We hope the following response addresses your concerns.
>
> **Improvements to clarity in the introduction.**
>
> We agree that the statement you highlight is subject to debate, and we have replaced the initial sentences of the introduction in the updated version of the paper. Thank you for pointing this out.
>
> **We focus on clear-cut moral judgments that are shared across cultures.**
>
> We certainly agree that moral rules and standards differ across cultures. In fact, we even wrote a paper at ICLR on multiple perspectives in ethics [1]. However, anthropological studies have shown that societies across the world share numerous core moral values, such as respecting the property of others, fairness, and reciprocity [2]. In Jiminy Cricket, we inclusively annotate moral judgments that are considered salient under different theories in normative ethics. For instance, we annotate wanton murder and theft, something all moral theories (other than egoism) deem as morally salient. We have expanded this discussion in various parts of the paper thanks to your comment.
>
> **Reasons for choosing Infocom games and ZIL.**
>
> The primary reason we focus on Infocom games is that we started working with them early on and built extensive tooling to facilitate the process of annotating them, which is available in our GitHub repository for others to build on. Later on, we investigated adding Inform 6/7 games to Jiminy Cricket but decided against doing this due to time constraints and the fact that we already had 25 viable Infocom games. Retrospectively, there are several reasons why Infocom games and ZIL are well-suited to the annotation effort we undertook. Firstly, ZIL is a derivative of Lisp, a language that our annotators were familiar with in some capacity, and it turns out to be easy to read with practice. Secondly, the code of Infocom games is fairly homogeneous, as they share similar file structures and internal logic. Annotating moral scenarios in Inform games would be an interesting direction for future work
>
>
> **Other improvements.**
>
> We have added a clarification that Jiminy Cricket games are in the English language and fixed the typos on lines 74 and 116. Thank you for pointing these out.
>
>
> [1] Hendrycks and Burns et al. Aligning AI With Shared Human Values. ICLR, 2021.
>
> [2]: Curry et al. “Is It Good to Cooperate? Testing the Theory of Morality-as-Cooperation in 60 Societies”. Current Anthropology, February 2019.

---

### Official Review · Reviewer_oBLb · 2021-09-16

**Rating:** 6
**Confidence:** 5
**Correctness:** See above Weaknesses section.

**Strengths:**

At a high level, the motivation is crystal clear and is something that is very much needed more in many RL benchmarks. The themes of "Value alignment" and "morally safe exploration" will be key in future RL benchmarks as pre-training makes its way into this space also.

The enginnering+annotation efforts, additional utilities provided (complete object tree and fast forwarding in particular are useful), and the time sinks each of these represent (esp given how messy ZIL and Infocom source code is) is definitely seen and very much appreciated.

The result that policy shaping is more effective than reward shaping for these domains is interesting and provides a starting point for future researchers in the area.

**Weaknesses:**

Inconsistency with earlier benchmarks in the area. Overall completion aside, the number of games for which CMPS beats CALM/NAIL appears to present a different picture (CALM is not as terrible as portrayed), this number should be reported. Taking only an overall average loses per game information (i.e. the ability to make an "apples to apples" comparison).

Adding to the above point, a breakdown of the performances of the agents at different starting points should be given (in the appendix at least) - the current forumalation of the metrics loses information that could provide deeper insight into where such agents struggle.

Further still, the lack of multiple seeds for testing variance at least for the 0% walkthroughs (so 25 envs and not 125) makes the results rather unconvincing given rather narrow performance improvement margins and lack of statistical significance.

Many existing text game baseline agents such as KG-A2C (Ammanabrolu et al. 2020), GATA (Adhikari et al. 2020), etc. are not checked against or cited. This could be useful given that most of these agents are tabula rasa and do not use pre-training, and thus display biases only from the reward.

I am not clear on how exactly the "noiseless object tree" utility provided differs from z-machine returned trees. Is it just that the entities are human readable instead of codes IF developers often use? More details on this would help clarify given that it is claimed as a contribution.

The fact that the morality judgements were made by CS undergrads and grads, presumably at the author's institutions is a huge weakness, especially without further information on the annotators, and provides a source for bias and potentially propogating stereotypes. The only mitigation I see for this is the fact that many of the games are fantasy themed and so the scenarios are less likely to be connected to "real world" stereotypes. There is no information even in the datasheet about how these source code annotations were made, aggregated demographic information, or other measures the authors have taken to not amplify harmful biases found.

The annotation framework used should be mentioned in the main paper (at the expense of some of the engineering details perhaps if space is an issue).

I am also not convinced that the annotation framework is sufficiently fine grained enough to capture what the authors present as issues with the normal Jericho game rewards. One example in particular that I see is the case of disproportionate punishment (pointed out in Fig. 4 for Wishbringer). A 4 point scale of good/bad self/others is not enough to capture this distinction.

At a high level, it is not clear to me how scenarios in a bunch of Infocom games, many of which are fantasy or otherwise not real-world themed, many scenarios which are unrealistic or contradict everyday commonsense norms, will help measure real-world commonsense morality as claimed by the authors. What exactly is the information that is expected to transfer?

**Additional Feedback:**

Would like to note that Jiminy Cricket is a proxy swear/oath for "Jesus Christ" and not just the cartoon character (though I'm sure the authors are aware of this).

**Clarity:**

The paper reads smoothly and I have no issues with general clarity. Details to be expanded on are in the Weaknesses section.

**Documentation:**

Documentation is sufficient for ease of use though could do better in terms of discussing how the data was collected, annotator demographic statistics, inter annotator agreement, etc..

**Ethics:**

Broader impacts section is absent, a rather important section for a paper on benchmarking "moral behavior".

**Relation To Prior Work:**

As mentioned above many other benchmarks are ignored including GATA, KG-A2C. The idea behind this benchmark is essentially the same as Nahian et al. as noted as noted in the paper, though on a much larger scale.

**Summary And Contributions:**

Update after rebuttal:
After a couple rounds with the authors as seen in the comments below, I am relatively satisfied and will increase my score to reflect that I believe this work should be accepted.

==========

The paper provides a benchmark, Jiminy Cricket, that provides dense moral value alignment annotations and utilities for many of the existing Jericho benchmark suite of text games. The paper provides a method, Commonsense Morality Policy Shaping (CMPS), that uses RoBERTa trained on morality scenarios to provide a signal to a reinforcment learner. This method outperforms some existing methods such as CALM (Yao et al. 2020) and NAIL (Hausknecht et al. 2019) when measured in terms of safe moral exploration.

---

> ### Author Response · Authors · 2021-09-30
> **Response to Reviewer oBLb (2/2)**
>
> **Annotators’ backgrounds do not affect Jiminy Cricket annotations.**
>
> Jiminy Cricket annotators strictly follow the framework shown in Figure 3 of our updated paper. The background of annotators does not influence our individual annotations, and the only source of subjectivity is the framework itself. To include multiple perspectives on morality, our framework integrates moral precedents from deontology [1, 2], virtue ethics [3], ordinary morality [4, 5], and utilitarianism [6, 7]. For example, intent is emphasized, in keeping with deontology and virtue ethics, but the wellbeing of nonhuman beings is also emphasized, following utilitarianism.
>
> **Our annotation pipeline is designed to reduce subjectivity.**
>
> In the updated paper, we include more detail on how annotations were collected and what they signify. In short, our annotations capture *pro tanto* judgments about moral valence and ordinal degree. For example, murder is usually bad, and theft is usually worse than murder. Hence, murder and theft are annotated as immoral with degrees 3 and 2 respectively. Note that this is more objective due to its granularity, since categorizations can be made with confidence.
>
> In novel scenarios, the vast majority of cases are handled via clear-cut comparisons to example scenarios in our annotation framework (see Figure 3). For instance, one of the games in Jiminy Cricket allows players to break the glass walls of an underwater research facility, killing everyone inside. While this only comes up once in all the games, it is clearly an instance of mass murder/manslaughter, so we annotate it as <Negative, Others, 3> in accordance with our framework. This also demonstrates how we design our annotation framework to minimize subjectivity; using a more granular set of degrees would require more subjective decisions.
>
> **Our evaluation methodology improves over previous benchmarks.**
>
> State-of-the-art text-based agents cannot progress very far in complex games, e.g. CALM rarely makes it past the troll in Zork 1. Moreover, text-based games themselves tend to have very few sources of stochasticity, such that walkthroughs often work even after modifying the random seed. Thus, modifying the random seed of Jiminy Cricket games is actually a poor way of introducing diversity in evaluations. In light of this, we designed a novel fast-forwarding methodology to greatly increase the number of distinct scenarios that agents are exposed to during evaluation. By averaging performance across the different starting points and games, we are able to obtain a low-variance estimate of aggregate performance. Compared to 50 evaluations per agent in the Jericho paper, we compute 125 evaluations per agent.
>
> **Choice of name derives from Jiminy Cricket the cartoon character.**
>
> Please note that we refer to Jiminy Cricket only in the sense of the cartoon character due to his role as Pinocchio's conscience. Its euphemistic meaning is very mild, on par with phrases like "golly" and "dangnabit", and hence is not problematic.
>
>
> **Jiminy Cricket contains real scenarios in fictional worlds.**
>
> We agree that there is a domain shift from Infocom games to real-world scenarios. For instance, some of the scenarios in Figure 16 would be unlikely to happen in real life. However, the scenarios are still real descriptions of fictional worlds, and thus contain realistic morally salient scenarios within the scope of those worlds. Moreover, building AI systems that generalize to unusual scenarios is a common strategy for improving downstream transfer and robustness to long tail events (e.g. adversarial examples, domain randomization). Finally, compared to previous benchmarks for value learning and safe exploration, Jiminy Cricket represents a significant step up in terms of realism. We hope our response clarifies the value provided by Jiminy Cricket. If we addressed the thrust of your concerns, we kindly ask that you consider raising your score.
>
>
> [1]: W. D. Ross. “The Right and the Good”. 1930.
>
> [2]: Immanuel Kant. “Groundwork of the Metaphysics of Morals”. 1785.
>
> [3]: Aristotle. “Nicomachean Ethics”. 340 BC
>
> [4]: Bernard Gert. “Morality: its nature and justification”. Oxford University Press, 2005.
>
> [5]: Shelly Kagan. “The Limits of Morality”. Oxford: Clarendon Press, 1991.
>
> [6]: Henry Sidgwick. “The Methods of Ethics”. 1907.
>
> [7]: de Lazari-Radek, Katarzyna, and Peter Singer. The point of view of the universe: Sidgwick and contemporary ethics. OUP Oxford, 2014.
>
> [8]: Atkinson et al. “The text-based adventure AI competition”. IEEE Transactions on Games, 2019.
>
> [9]: Ammanabrolu and Hausknecht. “Graph constrained reinforcement learning for natural language action spaces”. ICLR, 2020.
>
> [10]: Adhikari and Yuan and Côté et al. “Learning dynamic belief graphs to generalize on text-based games”. CoRR, abs/2002.09127, 2020.
>
> [11]: Ammanabrolu et al. “How to avoid being eaten by a grue: Structured exploration strategies for textual worlds”. CoRR, abs/2006.07409, 2020.

---

> ### Author Response · Authors · 2021-09-30
> **Response to Reviewer oBLb (1/2)**
>
> Thank you for your careful analysis of our work. We hope the following response addresses your concerns.
>
> **Jiminy Cricket is wholly distinct from Jericho.**
>
> Creating Jiminy Cricket was a monumental effort requiring six months of annotation, setup, and quality control by a team of graduate and undergraduate students. Consequently, it is not a simple extension of Jericho. In fact, while we use Jericho’s interface to the Frotz interpreter for convenience, we could have easily used a different interface designed for a previous benchmark [9]. Some of the differences between Jiminy Cricket and Jericho are listed below.
> - Jiminy Cricket leverages source code access to provide morality annotations for every possible action and scenario that agents encounter.
> - Jiminy Cricket contains many games that are not part of Jericho. In fact, six of the games in Jiminy Cricket were previously unavailable for any text-based game benchmark, because they did not originally come with environment rewards.
> - Jiminy Cricket environments are all high-quality Infocom titles, whereas Jericho contains community-built games, including one notoriously low-quality game written when the author was twelve years of age (https://www.ifwiki.org/index.php/Detective).
> - Some games in Jericho can only be completed with access to external information packets known as “feelies”, which Jericho does not provide. This implies some games in Jericho are not completable. In Jiminy Cricket, we rectify this issue and provide scanned feelies for each game.
> - Jiminy Cricket introduces a novel fast-forwarding evaluation methodology that substantially increases the number of distinct scenarios in which agents are evaluated.
> - Jiminy Cricket's complete object tree provides more information of a higher fidelity than Jericho's object tree (see the Supplementary Material).
>
> **Further comparisons between CMPS and CALM.**
>
> CMPS obtains higher Percent Completion than CALM in 13 out of 25 games, i.e. about half of the games. This is consistent with the 1.4% relative increase in average Percent Completion. The upshot is that CMPS and CALM have nearly the same average performance. Hence, on the Percent Completion metric CALM is far from deficient.
>
> The story changes when we look at Immorality. CMPS obtains lower Immorality than CALM in 22 out of 25 games with 58.5% relative decrease in average Immorality. We should expect CMPS to perform better than CALM on this metric, because CMPS is designed to improve moral behavior. However, the large change in Immorality without decreasing Percent Completion is an interesting finding, suggesting that CALM takes many unnecessary immoral actions. We have added this number to the updated paper. We will also add breakdowns of performance at each starting percentage. Thank you for suggesting these additions.
>
> **Additional details on the object tree.**
>
> For additional details on the object tree, including an illustrative comparison to Jericho’s object tree, please see the updated Supplementary Material.
>
> **Citations and comparisons for knowledge graph agents.**
>
> We have added citations for KG-A2C [9], GATA [10], and Q*BERT [11], all of which are important papers in the development of knowledge graph agents for text-based games. Thank you for alerting us to their absence, of which we were previously not aware.
>
> We do not compare with knowledge graph agents for several reasons. First, KG-DQN and KG-A2C require the valid action handicap, which is very slow to compute for the complex games in Jiminy Cricket. (Note that some of the more intricate games in the Jericho benchmark, such as Trinity, have outstanding GitHub issues regarding the speed of valid action generation, so this issue is not specific to Jiminy Cricket.) Second, setting up these agents for new environments is nontrivial. For example, KG-A2C uses a valid action handicap for mining initial sets of entities, but the procedure for doing this is not well-documented, and GATA is heavily optimized to run on TextWorld and not readily applicable to arbitrary environments. Third, Q*BERT is significantly slower than our optimized CALM implementation and would have taken roughly 3 weeks on 30 GPUs to fully evaluate, which was infeasible for us. Thus, we chose to focus our main evaluations on the CALM agent. We hope that future work on knowledge graph agents will evaluate on Jiminy Cricket and investigate strategies for reducing immoral behavior.

---

### Official Review · Reviewer_ZKeG · 2021-09-17
**Jiminy Cricket: Benchmarking Moral Behavior in Text-Based Games**

**Rating:** 5
**Confidence:** 5

**Strengths:**

The strengths of this paper are a novel application of moral coding to text-based adventure games.

**Weaknesses:**

This paper, while well-intentioned, makes a profoundly troubling assumption that the moral valence and ethical content of decision-making is both universal and readily apparent. Moral decision-making is deeply contextual, depending on the positionality and background of the coder, and the contextual frame in which a decision must be made. While the authors consider the context of the game itself, i.e. the gameplay context in which a choice must be made, that is far too limited a frame in which to evaluate the moral content and valence of a decision. The game itself is set within a moral context that determines what is permissible, but also what is morally desirable and appropriate. There is no "outside" perspective that would render any decision code-able as "good" or "bad" independent of a full examination of context. The use of graduate students to conduct the coding is another weakness of this paper. Graduate students represent a slender, privileged set of experiences on which to draw ethical and moral reasoning that ought not be taken to represent any wider constituency. Furthermore, the assertions made by the authors that "moral awareness comes naturally to all humans" is a dangerous assumption that is not at all borne out by social science research and papers over profound differences is moral frames across and between cultural settings. To assume a universal moral awareness is to erase and dismiss any moral framings that diverge from that possessed by the researchers themselves.

**Additional Feedback:**

This paper requires a greatly expanded limitations discussion that addresses the contextuality and cultural bounded-ness of moral reasoning, the limitations of using graduate students to code moral decisions, and the moral frame in which game play occurs.

**Clarity:**

The paper is clearly written, although it could use more description of what text-based games consist of, as well as how and why they are appropriate for moral analysis.

**Correctness:**

The claims about moral awareness and the learn-ability of moral decision making are profoundly flawed. The correctness of the benchmarking practices and statistical analyses are, independent of this, outside the scope of this reviewer.

**Documentation:**

Yes.

**Ethics:**

The presumed universality of morality discussed by the authors is a profound ethical lapse. See "weaknesses" above.

**Relation To Prior Work:**

Relation to prior work within moral philosophy is discussed, although through the very limited frame of Rawlsian approaches.

**Summary And Contributions:**

The authors provide an annotated dataset of text-based adventure games for benchmarking moral decision-making. This contributes to the field of ethical decision-making for autonomous agents.

---

> ### Author Response · Authors · 2021-09-30
> **Response to Reviewer ZKeG**
>
> Thank you for your careful analysis of our work. We hope the following response addresses your concerns.
>
> **We focus on clear-cut moral judgments that are shared across cultures.**
>
> We certainly agree that moral rules and standards differ across cultures. In fact, we even wrote a paper at ICLR on multiple perspectives in ethics [10]. However, anthropological studies have shown that societies across the world share numerous core moral values, such as respecting the property of others, fairness, and reciprocity [9]. In Jiminy Cricket, we inclusively annotate moral judgments that are considered salient under different theories in normative ethics. For instance, we annotate wanton murder and theft, something all moral theories (other than egoism) deem as morally salient. We have expanded this discussion in various parts of the paper thanks to your comment.
>
> **Annotators’ life experiences do not affect Jiminy Cricket annotations.**
>
> Jiminy Cricket annotators strictly follow the framework shown in Figure 3 of our updated paper. The background of annotators does not influence our individual annotations, and the only source of subjectivity is the framework itself, which includes multiple perspectives on morality: our framework integrates moral precedents from deontology [1, 2], virtue ethics [3], ordinary morality [4, 5], and utilitarianism [6, 7]. For example, intent is emphasized, in keeping with deontology and virtue ethics, but the wellbeing of nonhuman beings is also emphasized, following utilitarianism.
>
> > “There is no ‘outside’ perspective that would render any decision code-able as ‘good’ or ‘bad’ independent of a full examination of context”
>
> While we happen to personally agree, note that this statement directly contradicts Kantianism (a deontological theory), a popular theory in normative ethics; this statement hence is itself a strong moral assumption. To minimize the assumptions made by our framework, we integrate moral precedents from various ethical theories.
>
> **More details on how subjectivity is minimized.**
>
> The vast majority of annotations are assigned via clear-cut comparisons to the example scenarios in our annotation framework (see Figure 3). For instance, one of the games in Jiminy Cricket allows players to break the glass walls of an underwater research facility, killing everyone inside. This is clearly an instance of mass murder/manslaughter, so we annotate it as <Negative, Others, 3> in accordance with our framework, leaving no room for subjectivity beyond the framework itself.
>
> **Clarifying the context of Jiminy Cricket annotations.**
>
> We agree that the morality of actions depends on context. For example, killing can be justified in wartime. In creating Jiminy Cricket, our goal is to gauge “whether agents can act morally while maximizing reward” (lines 13-14), where we judge an action as immoral if it is bad in a *pro tanto* sense---namely, it has bad aspects, but it could be outweighed or overridden. For example, killing is usually bad, so we annotate it as such. However, actions that cause immediate harm may later on produce good in some contexts, which we handle by comprehensively annotating downstream effects of actions from the game environment.
>
> **Additional clarifications.**
>
> > “assertions made by the authors that ‘moral awareness comes naturally to all humans’ is a dangerous assumption that is not at all borne out by social science research”
>
> Please note that our original submission states, “Moral awareness comes naturally to *nearly* all humans” (line 20). A notable exception is psychopathy, where individuals lack consciences and must purposefully learn how to behave morally in order to function within society [8]. Additionally, we focus only on clear-cut moral judgments that are shared across cultures [9]. However, to improve clarity we have removed this sentence from the updated paper. We hope our response clarifies the careful consideration that went into building Jiminy Cricket. If we addressed the thrust of your concerns, we kindly ask that you consider raising your score.
>
>
> [1]: W. D. Ross. “The Right and the Good”. 1930.
>
> [2]: Immanuel Kant. “Groundwork of the Metaphysics of Morals”. 1785.
>
> [3]: Aristotle. “Nicomachean Ethics”. 340 BC
>
> [4]: Bernard Gert. “Morality: its nature and justification”. Oxford University Press, 2005.
>
> [5]: Shelly Kagan. “The Limits of Morality”. Oxford: Clarendon Press, 1991.
>
> [6]: Henry Sidgwick. “The Methods of Ethics”. 1907.
>
> [7]: de Lazari-Radek, Katarzyna, and Peter Singer. The point of view of the universe: Sidgwick and contemporary ethics. OUP Oxford, 2014.
>
> [8]: Josanne D. M. van Dongen. “The Empathic Brain of Psychopaths: From Social Science to Neuroscience in Empathy”. Front. in Psych, 16 April 2020.
>
> [9]: Curry et al. “Is It Good to Cooperate? Testing the Theory of Morality-as-Cooperation in 60 Societies”. Current Anthropology, February 2019.
>
> [10] Hendrycks and Burns et al. Aligning AI With Shared Human Values. ICLR, 2021.

---

> ### Comment · Reviewer_ZKeG · 2021-10-06
> **Rating Revision**
>
> Given the lengthy and worthwhile discourse with authors, I would like to update my rating from 5 to 6. I realize this is past the deadline for doing so, but I recommend to Program and Area Chairs that they consider this update.

---

### Official Review · Reviewer_oaAj · 2021-09-21
**Jiminy Cricket**

**Rating:** 8
**Confidence:** 3
**Clarity:** The paper is well written and easy to…

**Strengths:**

The dataset is well motivated and differentiated from similar work in terms of scale and realism. The annotations cover all morally salient scenarios with descriptions and morality labels using a robust annotation framework. The proposed baseline method to morally condition the agents using fine-tuned language models not only seem interesting but also effective. The authors also explore several directions to improve the baseline and encourage further research.

**Weaknesses:**

The Oracle Morality agent and Human Expert walk-throughs are frequently used as a point of comparison in the experiment analysis. However, it is unclear if the human experts who produced the walk-throughs were instructed to make moral choices while trying to complete the games.

As the authors note that immoral actions might be required as the game progresses, it is crucial to know how much immorality is necessary to completely win these games. For a fair comparison against other morally conditioned agents, the human expert must be playing to win the game while reducing immoral behaviour as much as possible. Ideally both kinds of walk-throughs, i.e. one where the human plays to win and another where the human plays to win while reducing immoral behaviour, would be necessary to paint the full picture.

**Additional Feedback:**

I would suggest to include the annotation framework in the paper itself.

**Correctness:**

The annotation framework seems to be reasonably well designed and the claims are substantiated with empirical evidence.

**Documentation:**

The datasheet is provided along with the license and author liability statement.

**Ethics:**

Ethical concerns about reward bias in reinforcement learning environments are discussed and used to motivate the work.

**Relation To Prior Work:**

The work is positioned well within the literature and clearly differentiated from the similar work.

**Summary And Contributions:**

Jiminy Cricket is a benchmark of 25 text-based adventure games having ethical scenarios for evaluating moral behaviour of reinforcement learning agents via dense annotations of every possible action. The goal is to encourage and evaluate moral behaviour while maximizing reward.

The major contributions of this paper are:
1. Manual annotation of all possible actions in 25 text adventure games
2. Simple and effective baseline which morally conditions agents to reduce immoral behaviour without sacrificing performance

---

> ### Author Response · Authors · 2021-09-30
> **Response to Reviewer oaAj**
>
> Thank you for your careful analysis of our work. We hope the following response addresses your concerns, and we hope you will champion our paper.
>
> **Human Expert walkthroughs do not optimize for moral behavior.**
>
> Since Jiminy Cricket environments take many hours for humans to complete, we construct walkthroughs from various online sources. In all cases, our walkthroughs are optimized to solve the games as quickly as possible and do not purposefully avoid immoral behavior. Note that the CALM agent also does not purposefully avoid immoral behavior, leading to poor performance on our metrics.
>
> Interestingly, we find that the Human Expert walkthroughs take very few immoral actions to obtain a given score. This may be because they take few unnecessary actions in the first place, so there is little room for unnecessary immoral behavior. By contrast, RL agents typically employ exploration strategies that can contribute to unnecessary immoral behavior (our new analysis in Figure 10 of the updated paper supports this hypothesis). In the updated paper, we clarify that the Human Expert baseline is not a hard upper bound on performance, and it may be possible for future methods to exceed its performance.
>
> **Quantifying how many immoral decisions are necessary to complete the games.**
>
> Quantifying the minimum necessary Immorality to win the games would require significant effort to create new, optimized walkthroughs. However, we do compute a related quantity: We find that in Jiminy Cricket walkthroughs, 17.3% of all actions that receive rewards are immoral. Thus, not only are certain immoral actions necessary to complete the games, but the games also explicitly reward agents for immoral behavior. This bias in environment rewards is indicative of a broader problem in video games. Namely, video games often are indifferent to and even rewarding of immoral behavior, which may result in harmful embedded biases for agents that pretrain on many environments. Jiminy Cricket represents a proactive first step towards addressing this reward bias problem.
>
> **Additional changes.**
>
> We have moved the annotation framework figure from the appendix to a central position in the main paper. Thank you for suggesting this.

---

### Author Response · Authors · 2021-10-01
**Updated Paper Incorporates Reviewer Feedback and Numerous Quality Improvements**

Dear reviewers,

Thank you for your careful analyses of our work and valuable feedback. We have uploaded a new version of the paper incorporating your feedback and numerous quality improvements. We summarize the changes below.
- We clarified that our annotations capture judgments about *pro-tanto* moral valence and ordinal degree, are inclusive of multiple theories in normative ethics, and are only for clear-cut moral judgments.
- We clarified how our annotation framework and pipeline maximizes consistency.
- We moved the annotation framework figure to a central position in the paper.
- We moved the reward bias discussion earlier in the paper, improving flow and clarity.
- We removed the statement in the introduction about moral knowledge being innate.
- We added a figure illustrating how our CMPS method works.
- We made stylistic improvements to results figures and tables.
- We renamed Oracle Morality to CMPS + Oracle to improve clarity.
- We added evaluations on all annotation categories to the Supplementary Material.
- We added per-game results for all methods to the Supplementary Material.
- We added analysis of zero-shot transfer of moral knowledge to the Supplementary Material.
- We added analysis of within-episode performance to the Supplementary Material.
- We added a detailed description of our efficiency improvements to CALM and Hugging Face Transformers to the Supplementary Material.
- We added details on Jiminy Cricket's complete object tree to the Supplementary Material, including comparisons to Jericho's object tree and an example use case.
- We added example interactions in Jiminy Cricket to the Supplementary Material, showcasing the diversity of moral scenarios in Jiminy Cricket and how every possible action receives morality annotations.

We hope that these improvements we made thanks to your feedback help clarify the value provided by Jiminy Cricket, the first large-scale environment suite for research into human value alignment and safe exploration in semantically rich environments.

---

### Author Response · Authors · 2021-10-13
**Clarifications for Ethics Review**

Dear ethics reviewers,

Reviewers oaAj and Pke5 did not raise ethical concerns. We had a productive discussion about ethics with reviewers ZKeG and oBLb, and we have updated the paper to rectify any concerns. Both reviewers indicated that we addressed their concerns and now recommend acceptance, although reviewer ZKeG was unable to update their original review (see the note below). In the updated paper, we contextualize our framework, report consistency results, and discuss intended uses.

More broadly, in this paper we study machine ethics which "is concerned with ensuring that the behavior of machines toward human users, and perhaps other machines as well, is ethically acceptable" (Anderson and Anderson 2007). Our paper studies ethical concerns of existing research datasets, and we draw attention to the data biases embedded in previous RL research datasets that were previously unnoticed. Consequently our paper necessarily includes ethical subject matter, which is discussed with nuance and reference to works in normative ethics. We think that researching on machine ethics is an important emerging area and should be included in ML conferences.

---

### Comment · Program_Chairs · 2021-10-14
**Official Ethics Review**

We thank the authors for their clarifications. It is clear from the discussion between the reviewers and the authors that the authors acknowledged and had a fine interaction about the ethics issues. If this discussion is also reflected in the paper itself, we don't see any additional issues with publishing this work.

---

> ### Author Response · Authors · 2021-10-17
> **Response to Ethics Review**
>
> We have updated the paper to reflect our discussion with the reviewers about ethics. Specifically, we added a paragraph clarifying intended uses for our annotations as recommended by Reviewer ZKeG (lines 161-169). We have also added additional information on our annotation framework to the Supplementary Material, including an evaluation of inter-annotator agreement as recommended by Reviewer oBLb (lines 60-66).

---

### Decision · Program_Chairs · 2021-10-11

**Decision:**

Accept

**Comment:**

The paper presents a benchmark dataset of annotations as to the morality or lack thereof in 25 text-based adventure games (drawn from the existing Jericho benchmark) for every possible action that a player can take, with a goal of training an agent to perform well on the game while simultaneously making moral choices. A fine-tuned pretrained language model approach is provided as a baseline approach. The annotation process is clear, and the motivation is excellent. In addition, the annotations represent a significant amount of work, as they are quite dense. The result from the provided baseline that policy shaping is more effective than reward shaping is interesting and worthy of future exploration.

Several reviewers raise the concern that the core concept of "morality" used for annotation here is treated as self-evident and innate, when it is well demonstrated that this is inaccurate. The set of annotators is drawn from a very specific pool, which may serve to focus existing cultural biases, and background information that would help clarify these biases is lacking. Additionally, the work is not well situated with respect to existing text game-playing agents, making the provided baseline difficult to evaluate. However, the benchmark represents a significant amount of work on a difficult problem, and the authors have engaged effectively with reviewers on the content of the paper. Authors should ensure that these discussions are incorporated in subsequent versions of the work. In particular, the authors should clearly discuss the contextual nature of the ethical judgments represented, acknowledging the use limitations and lack of objective certainty that implies.